# Improving Cathode Testing with a High-Gradient Cryogenic Normal Conducting RF Photogun

**Gerard Emile Lawler** [1,*], **Fabio Bosco** [1,2], **Martina Carillo** [2], **Atsushi Fukasawa** [1], **Zenghai Li** [3], **Nathan Majernik** [3], **Yusuke Sakai** [1], **Sami Tantawi** [3], **Oliver Williams** [1], **Monika Yadav** [1] and **James Rosenzweig** [1]

1   Department of Physics and Astronomy, University of California, Los Angeles, 470 Portola Plaza, Los Angeles, CA 90095, USA
2   Department of Basic and Applied Sciences for Engineering, La Sapienza University of Rome, 00185 Rome, Italy
3   SLAC National Accelerator Laboratory, 2575 Sand Hill Rd., Menlo Park, CA 94025, USA
*   Correspondence: gelawler@physics.ucla.edu

**Abstract:** Future electron accelerator applications such as X-ray free electron lasers and colliders are dependent on significantly increasing beam brightness. With the observation that linac beam manipulation's best preservation of max brightness is at the cathode, we are incentivized to create an environment where we can study how to achieve the highest possible photogun brightness. In order to do so, we intend to extract beams from high-brightness photocathodes with the highest achievable accelerating gradients we can manage in a klystron-powered radiofrequency (RF) photogun. We utilize here cryogenic normal conducting cavities to achieve ultra-high gradients via limitation of breakdown rates (BDR). The low temperatures should also reduce cathode emittance by reducing the mean transverse energy (MTE) of electrons near the photoemission threshold. To this end, we have designed and produced a new CrYogenic Brightness-Optimized Radiofrequency Gun (CYBORG) for use in a new beamline at UCLA. We will introduce the enabling RF and photoemission physics as a primer for the new regime of high field low temperature cathodes we intend to enter. We further report the current status of the beamline commissioning, including the cooling of the photogun to 100 K, and producing 0.5 MW of RF feed power, which corresponds to cathode accelerating fields in the range of 80–90 MV/m. We further plan iterative improvements to both to 77 K and 1 MW corresponding to our ultimate goal >120 MV/m. Our discussion will include future beamline tests and the consideration of the initial realization of an ultra-high-gradient photoinjector concept.

**Keywords:** high-gradient cavities; cryogenics; photoemission; high-brightness beams





## 1. Introduction

One of the most promising directions in the development of future electron linear accelerators is improving their performance for high-impact applications by increasing beam brightness. Higher beam brightness is associated with the improved functionality of existing machines for scientific users and reducing engineering costs, allowing for more accessibility to future machines with lower initial investment. Some of the more notable applications include X-ray free electron lasers (XFELs) [1–3], linear colliders [4,5], ultra-fast electron diffraction [6–8], inverse Compton scattering [9,10], and compact medical linacs [11].

The size of linear accelerators can be kept in check with bright beams produced with low mean transverse energy (MTE) high quantum efficiency (QE) photocathodes [12–14] and accelerated in higher-gradient cavities [15]. This becomes especially relevant as the march towards arbitrarily large machines becomes less and less feasible. The primary illustrative motivational scaling law we refer to is the beam brightness produced in a radiofrequency (RF) photogun as follows:

$$B_{e,b} \approx \frac{2ec\varepsilon_0}{k_B T_c}(E_0 \sin \phi_0)^2 \tag{1}$$

where $e, c, k_B$, and $\varepsilon_0$ are the relevant fundamental constants, $T_c$ is the thermodynamic temperature of the cathode, $E_0$ is the launch field magnitude, and $\phi_0$ is the accelerating field phase. The simplified 1D case is not a complete picture of photogun physics but is a good example of the sort of beneficial effects we hope to utilize to obtain our desired beam brightness increase [16]. We can see a squared dependence on the launch field at the cathode and inverse dependence on temperature when near threshold photoemission is achieved, thus providing a strong incentive to increase brightness via high gradient operation and potential advantageous low temperature cathode emission behavior. It is then advantageous to consider a machine which can provide both high cathode fields and low temperatures taking advantage of two phenomena to improve brightness as per Equation (1).

Many of these ideas are epitomized within the ultra-compact x-ray free electron laser (UCXFEL) concept [1]. The UCXFEL and associated TopGun photoinjector use extremely high gradients enabled in RF cavities at cryogenic temperatures to improve beam brightness and reduce linac length. It has long since been shown that the cryogenic operation of normal conducting radiofrequency (NCRF) cavities significantly reducing the breakdown rate (BDR) [15]. BDR is often the value most limiting high gradient cavity operation so indeed field in excess of 500 MV/m have been observed in pillbox structures [17]. Utilizing such enhanced fields at cryogenic temperatures, one could achieve unprecedented high accelerating gradients of 240 MV/m realistically with existing cooling technologies.

When combining high gradients in the photogun and subsequent accelerating structures with high-performance photocathodes, we can expect to shrink a kilometer-scale linac (in this case for reference the LCLS at SLAC is useful for comparison) down to tens of meters. In order to keep thermal loads from RF pulse heating manageable, high shunt impedance re-entrant cavities are used. The beam dynamics in these novel cavity geometries have been studied extensively [1,18–20]. The Cool Copper Collider ($C^3$) collider concept provides another realistic initiative that can utilize cryogenic RF and photoemission principles to obtain improved performance on a more affordable scale (especially when considering other linear collider concepts like the ILC) [4].

For the most part, the understanding of the enabling phenomena here mentioned are based largely on empirical studies and the basic physics is not completely understood. This is especially true when we consider the multi-scale physics question of the breakdown phenomenon [21–23]. The behavior of novel semiconductor cathodes with complex band structures has more theoretical underpinnings but is not significantly explored in certain extreme photoinjector regimes [24]. To obtain an idea of cryogenic cathode behavior, we can consider the effect of cathode temperature on the intrinsic emittance of a simple metallic cathode via the expressions from Vecchione et al. [25]. The results are plotted in Figure 1. We calculate the intrinsic emittance at the cathode as a function of excess energy for a number of temperatures for a given spot size of 75 μm. We can see that in the absolute limit of near-threshold photoemission where thermodynamic temperature dominates, we would expect around a factor of three decrease in intrinsic emittance from 295 K down to 45 K. The minimum temperature of 45 K is considered as it is the temperature for which the highest gradient fields have been measured in the previous NCRF cavity study [17]. Subsequently this value was used in the UCXFEL study as the best-case scenario. For our case, we further consider that the excess energy is an issue of tuning the photoemission laser to a value near the threshold of photoemission. It then becomes more illuminating for our consideration to look at temperature dependence for certain incident laser parameters, which we also show in Figure 1.

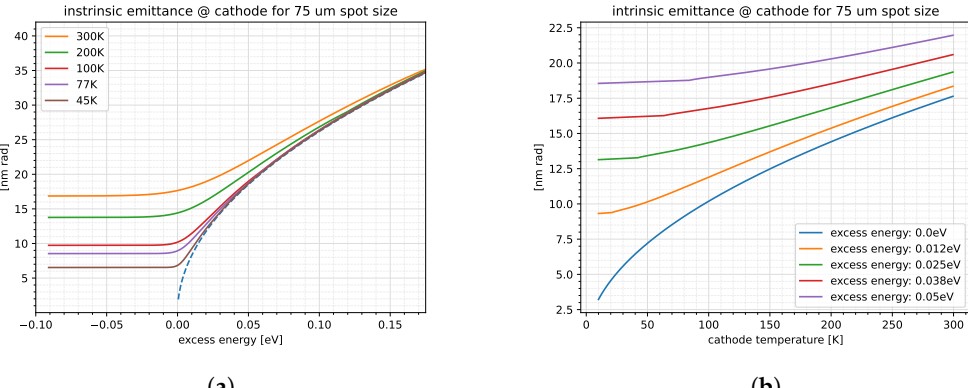

(**a**)            (**b**)

**Figure 1.** Cryogenic predictions of intrinsic emittance from metallic cathodes calculated via previous derivations [25]. (**a**) Intrinsic emittance as function of excess energy shown for relevant low-temperature curves. The dashed curve represents the asymptotic solution. (**b**) Intrinsic emittance as function of temperature for set excess energy.

## 2. Materials and Methods

We recognize here that improving beam brightness necessitates the development of new laboratory facilities. The goal here is to develop a more in-depth understanding of the underlying physics associated with photoemission in high gradient fields at cryogenic temperatures with a compact machine in a laboratory setting. To provide a frame of reference, it is useful to consider the state-of-the-art in terms of high-gradient and low-temperature cathode testing. In Table 1, we compare certain relevant existing (both commissioned and developing) photoguns that occupy a similar space as our intended exploration. The guns considered can accommodate the testing of normal conducting cathodes and have high-peak cathode fields ($\geq$10 MV/m) or cryogenic cathode temperatures ($\leq$120 K). The columns are organized roughly with the highest gradients on the left going to coldest temperatures on the right in order to reflect the general trend of lower-temperature guns possessing lower-peak cathode fields.

One can see that the highest-gradient cryogenic tests have been in superconducting radiofrequency (SRF) and DC guns with fields significantly lower than what even a room temperature NCRF photogun test bed can provide. Furthermore, SRF cavities are significantly more difficult to fabricate [26]. Existing high-gradient NCRF photoguns used for cathodes, however, cannot be cooled to the cryogenic temperatures of interest to advanced cathode research. Furthermore, due to the reduction of breakdown, cryogenic NCRF guns can operate at still higher fields. The state of the art includes additional photoguns that have been designed and undergone preliminary testing, which are not included in the table as they are not currently integrated into beamlines for cathode testing [27–31].

To address the gap in cathode testing capabilities, we have designed a CrYogenic Brightness-Optimized Radiofrequency Gun (CYBORG) for producing high fields at low temperatures in order to study novel cathodes. The gun geometry is shown in Figure 2. In order to evaluate the performance of CYBORG and its use in a cryogenic photoemission beam line, we consider a number of different forms of analysis for our study. We must consider theoretically the importance of brightness; the cryogenic behavior of electron emission; the issue of preserving high brightness from the cathode; and how to increase accelerating gradients by way of reducing BDR. The CYBORG beamline is being commissioned in two phases: Phase 1 involves conditioning the gun for high gradients at cryogenic temperatures and low emittance measurements of cryogenic copper cathodes; Phase 2 will include a load-lock for the insertion of advanced cathodes into the gun.

**Table 1.** Parameters for a selection of state-of-the-art guns for novel cathode testing. Certain similar guns are grouped together where appropriate.

| Photoguns | FERMI [32] | PEGASUS [7,33] | PITZ [34,35] | HZDR [36] /HZB [37] | Cornell [38] /ASU [39] | BNL [40,41] |
|---|---|---|---|---|---|---|
| Cavity type * | NCRF | NCRF | NCRF | SRF | - | SRF |
| Cavity geometry * | 1.6 cell pillbox | 1.6 cell pillbox | 1.5 cell pillbox | 1.5 cell elliptical | - | Quarter wave |
| Cathode assembly | Demountable Cu backplate | Demountable Cu backplate + load-lock | Demountable Cu backplate + load-lock | Cryogenic load-lock | Cryogenic load-lock | Cryogenic load-lock |
| Design frequency | 2.998 GHz | 2.856 GHz | 1.3 GHz | 1.3 GHz | DC | 0.113 GHz |
| Peak cathode field | 125 MV/m | 120 MV/m (Cu backplate) | 60 MV/m | 15–20 MV/m | 10 MV/m | 10–15 MV/m |
| Min cathode T | ≥room T | ≥room T | ≥room T | 80 K | 35 K | 2 K |

\* Only relevant for RF guns.

We further note that improved understanding of basic material physics and the phenomena of breakdown are necessary especially at cryogenic temperatures. To this end, we take an integrated iterative approach to our simulations and experiment where our suite of simulations and theory are continually improved as our gun is commissioned and empirical measurements are made of materials and RF properties. Pragmatically we consider a number of figures of merit from beam dynamics simulations (using GPT), electromagnetic simulations (using CST), thermo-mechanical simulations (using SolidWorks and CST), and various measurements of certain subsystems. Then using the operational parameters of the space of existing guns for basic cathode experiment we can place the CYBORG test bed in the context with the state of the art.

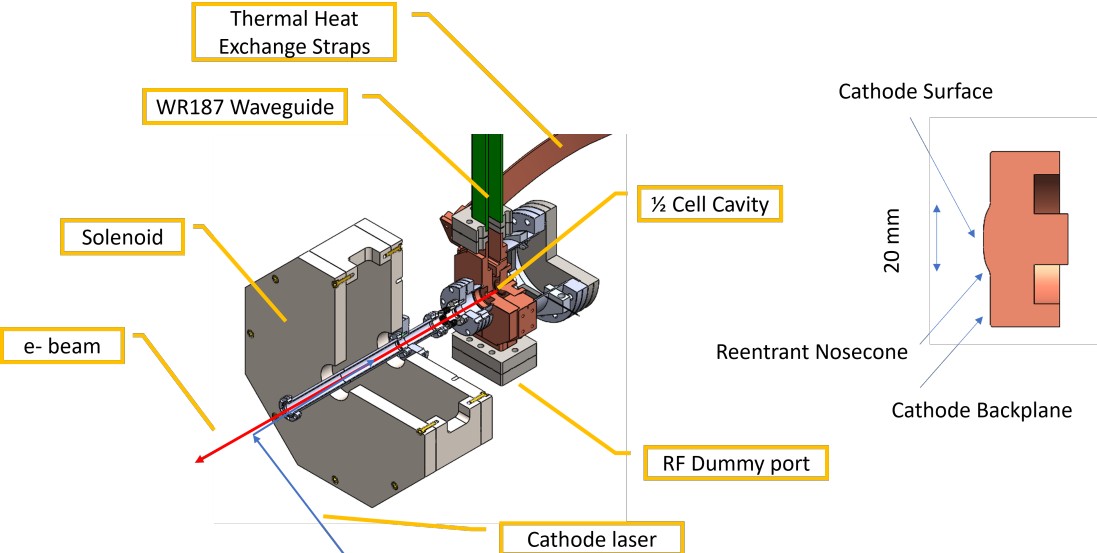

**Figure 2.** CYBORG configuration for Phase 1 measurements (**left**) Cutaway showing half-cell cavity and select subsystems. Note that the solenoid is room temperature and outside of the cryostat. (**right**) Cross-section of the Phase 1 polished copper cathode.

## 2.1. CYBORG RF Theory and Simulation

As our primary photogun theory, we perform an extensive suite of RF simulations to obtain predictions of the low-temperature figures of merit for our re-entrant high shunt impedance cavity. Figure 3 shows the gun cavity design with electric field magnitude

plotted using 1 J of stored energy for simulation purposes. Originally designed with a racetrack feed for field symmetry, one side (the left in the Figure 3) was instead turned into a dummy port with a choke to allow for RF probe measurements and to limit the space needed within the gun cryostat. One can further observe the RF waveguide coupler that feeds the cavity on the right and also the peak cathode fields concentrated on the re-entrant nosecone.

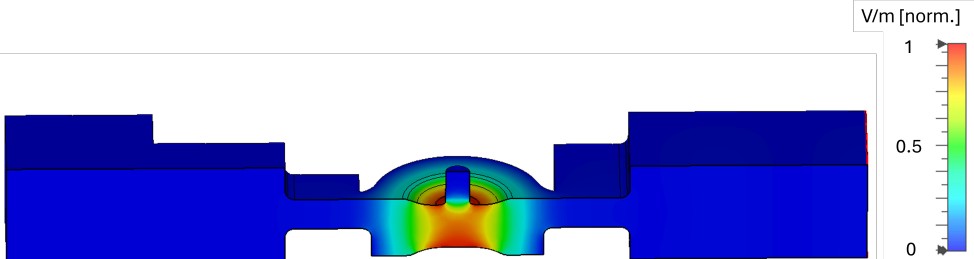

**Figure 3.** Cavity fields normalized to the peak value within the gun cell showing the peak fields on the cathode side re-entrant nosecone and dummy port after a choke in the left coupler.

The primary figures of merit here are quality factor $Q_0$, waved to cavity coupling $\beta$, shunt impedance $Z_{shunt}$, filling time $\tau$, RF power needed to maintain a given accelerating voltage, and RF pulse heating [42]. The last of which, the RF pulse heating, is especially important when we consider temperature stability in the form of various thermomechanical simulations. The simulations are relatively straight forward to perform in multiphysics software given appropriate boundary conditions and knowledge of electromagnetic fields but they become more challenging when temperature dependence is introduced. Material properties such as electrical and thermal conductivity and coefficient of thermal expansion change as a function of temperature. Furthermore, for cryogenic temperatures at intermediate values (specifically in our range of operational interest of 45–100 K) they do not necessarily have convenient asymptotic values as they would in the high temperature and 0 K limits [43].

In order to accurately incorporate explicit temperature dependence we need to include more nuanced theoretically founded calculations of bulk material properties such as electrical and thermal conductivity and coefficients of thermal expansion as well as surface physics in RF cavities. We begin with the Bloch–Gruneisen model to explicitly calculate bulk electrical resistivity (simply the direct inverse of conductivity) in terms of temperature [44]. A direct calculation of this expression leads to vanishing resistivity at 0 K so we further incorporate an additive factor called the residual resistivity ratio (RRR), which we can use to parameterize certain unknown bulk properties like purity, crystal grain structure, etc., all of which increase the 0 K resistivity. Thus we have the following

$$\rho(T, RRR) = A \left( \frac{T}{\Theta_R} \right)^n \int_0^{\Theta_R/T} \frac{t^n}{(e^t - 1)(e^t - 1)} dt + C(RRR) \tag{2}$$

where $n = 3$ is sufficient for transition metals, $T$ is temperature, $A$ is a metal-dependent constant, and $\Theta_R$ is sufficiently close to the Debye temperature to warrant its use here for our case. For completeness, we can compare these bulk predictions to curves and data published by the National Institute of Standards and Technologies (NIST). In general, based on previous work the high-purity oxygen-free copper that is used for cavity manufacture, the RRR is in the range of several hundred with cavities meant for high-power BDR testing [45]. We plot in Figure 4 some example values for a range of RRR values as a function of temperature.

For the thermal conductivity and coefficients of thermal expansion we use direct NIST data fitted to high degree logarithmic polynomials for accuracy down to 4 K. Furthermore for RF cavities an additional consideration must be made for low temperature operation as we enter the anomolous skin effect (ASE) regime, where the mean free path length of an

electron in the bulk and the skin depth of the RF become comparable, leading to increases in the surface resistivity $R_s$ [46–48]. We can compute the relevant length scales of these phenomena and develop more nuanced theories than the Reuter and Sondheimer theory based on thin film physics [49,50]. We can further use the ASE and extended ASE $R_s$ to compute effective bulk properties from an effective skin depth to introduce this theory with simplicity into existing multiphysics simulations. With these temperature dependent $R_s$ curves we are free to perform desired simulation sweeps for arbitrary temperatures by introducing the modified material properties which can then be compared to the explicitly measured experimental values.

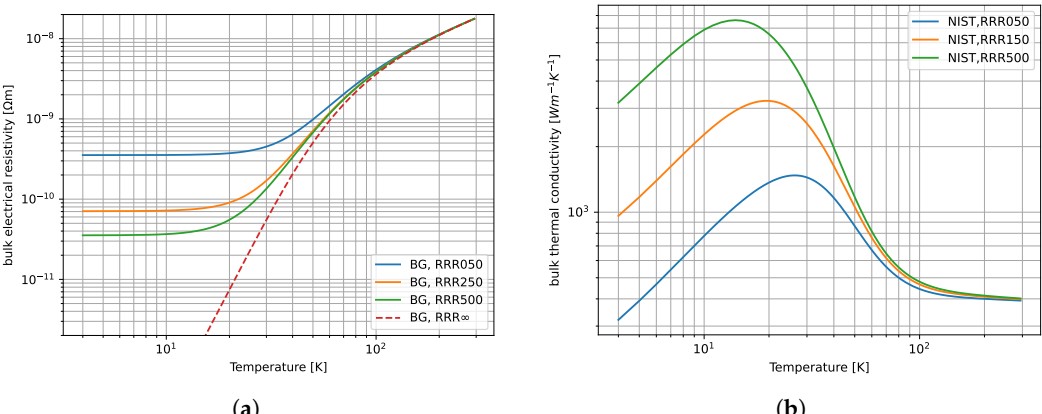

**Figure 4.** Bulk material properties for various RRR values computed for the oxygen-free copper similar to that used to manufacture CYBORG. They are computed from Equation (2) and NIST polylogarithmic fits. These values, when combined with experimentally measured quantities when possible, provide the basis for our gun and beamline characterization. (**a**) Bulk copper resistivity (**b**) Bulk copper thermal conductivity.

Many of the RF figures of merit are dependent directly on the RF source properties. Due to our cryogenic needs, the frequency of operation for our gun is in C-band to limit thermal requirements [19]. Since C-band is a less common frequency range for lab infrastructure we developed and commissioned a custom klystron modulator with pulse-forming network (PFN) built in-house at UCLA for use with an existing Thales C-band klystron tube. The tube itself uses hybrid infrastructure from a decommissioned SLAC XK5 klystron where possible, including the pulse tank. Where possible for our theoretical predictions, we use measurements of characterized infrastructure or otherwise circuit simulations developed in Spice. The main measurable quantities we focus on are the unloaded quality factor $Q_0$, resonant frequency $f_0$, coupling $\beta$, and the filling time $\tau$. These quantities are measured in low-power tests by the $S11$ minimum found with a network analyzer and by analyzing forward and reflected pulses in high power tests.

### 2.2. Photoemission and Beamline Simulations

A direct examination of the theory of photoemission is limited within the scope of the studies here to the simple ideal metal results shown in Figure 1. For more advanced semiconductor cathodes, of interest to certain high brightness applications, we consider the most recent experimental results where possible. We use these results to inform the initial beam parameters for our photoemission model for the sake of temperature-dependent beamline simulations.

Magnetic optics, such as the existing quadrupoles, have been characterized with a newly developed 3-axis high-precision measurement gantry for 3D field measurement. This becomes important for a low-emittance beamline since any spurious increase from for example fringe fields would destroy the very low cryogenic emittance [12,51,52]. In this context, a novel multi-start foil wound solenoid was designed, which is further detailed in the following publication [18]. Progress is ongoing, since the technology is quite advanced.

The current challenge mainly involves the slow progress of manufacturing insulated copper foils of sufficient quality for assembly. For the near term, we have manufactured a conventional coil wound solenoid around the optimized yoke geometry. The existing stepping stone solenoid is intended for functional near-term measurements, so has been characterized using the gantry probe. These measured fields can then be imported into our GPT models for simulation.

## 3. Results

### 3.1. Cooling and Gun Temperature Stability

The first step of our analysis was to measure temperature stability of the gun as a static load. We assessed a number of heat leaks to incorporate into our thermal simulations including most significantly the straight steel waveguide section. We use this piece as a first iteration of a thermal break between the lowest-temperature CYBORG can achieve and the room-temperature waveguide window outside the cryostat. Steel has one of the higher thermal resistivity values for a metal, so it is the simplest solution.

Using the methodology we established in the previous section, we can simulate the transient thermomechanical behavior during the cooldown cycle of the cryogun. We compute these results in the case where gun rep rate is 1 Hz. This implies first of all that internal RF pulse heating is insignificant in our case such that we can simplify our cool-down simulation. We then compare simulations to temperature measurements of the gun and cryocooler within the existing gun cryostat environment. These measurements and simulations are show in Figure 5.

Currently our gun temperature is limited to 95 K given the significant heat leaks from the waveguide, beam pipe, and black body radiation from the cryostat walls. The design goal of 77 K is not required for the Cu cathode studies of Phase 1 and 100 K is sufficient for now. The 95 K temperature is well within the regime of interest for initial copper MTE improvements given our laser parameters. The cooling simulation agrees well with the experiment. In addition, a suite of transient thermal simulations is being developed to further quantify the effects of RF pulse heating.

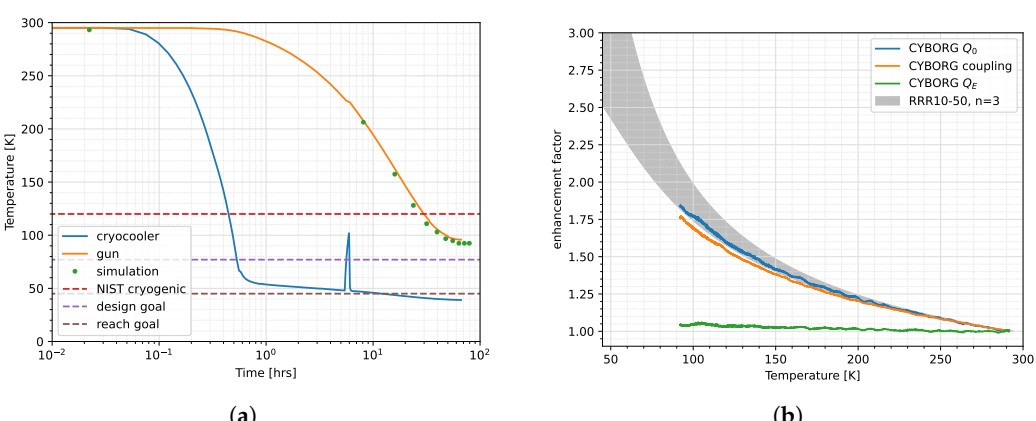

(**a**)          (**b**)

**Figure 5.** CYBORG cool-down data measured to 95 K. (**a**) Photogun and cryocooler temperatures as function of time compared to design goals and simulation. The spike in cryocooler temperature was an intentional cooler shut down to test heating and measure strap conductance. (**b**) RF FoM measurements compared to simulation and theory via Equation (2). Note differences between the normalized coupling, and $Q_0$ implies that $Q_E$ changes as a function of T.

Phase 2 will begin with more precise near-threshold laser illumination for semiconductor cathode measurement necessitating cooling to 77 K. Original plans call for an outer cold shield cooled to 150 K, which will significantly reduce gun temperature. This will be in addition to the existing inner shielding made of multi-layer insulation (MLI). The outer thermal shield is currently being fabricated. The stretch goal of 45 K is included only as a means to orient towards a future crycooled photoinjector and is not a working point for

CYBORG with existing infrastructure. Additional novel thermal insulation techniques and cooling power are necessary to achieve this temperature. Part of the function of CYBORG is to test cryocooler integration on a beamline (i.e., vibration reduction, alignment, etc.) and RF cavity figures of merit (i.e., temperature-dependent properties, etc.) as a stepping stone to a UCXFEL photoinjector, which is designed to operate at 45 K, so the inclusion of theoretical performance is useful [53].

*3.2. RF FoM with Temperature Dependence*

During the cool-down, we also made low-power RF measurements via an RF antenna on the waveguide ≈4.5 m upstream from the gun. The results are shown compared to theory in Figure 5. The values of the unloaded quality factor $Q_0$ and coupling $\beta$ are measured down to 95 K. They are plotted here as quantities normalized to the value at room temperature and we thus refer more accurately to the plotted quantities as enhancement factors. The value of the $Q_0$ enhancement factor as a function of temperature falls within the theoretical range of RRR values between 10 and 50.

The oxygen-free copper used in CYBORG manufacture has a nominal literature value of RRR between 100 and 500, and there is no direct reason to think the copper material properties have been altered. Based on an inspection of the cavity, the surface does not seem to be of degraded quality. Indeed, a pillbox cavity manufactured with the same materials and brazed in the same oven using the same procedure was tested and the expected higher $Q_0$ enhancement factors were observed [50]. A certain reduction in the effective RRR was expected based on previous studies [54]. It is important to note that this RRR is calculated not directly from conductivity measurements and instead from measured $Q_0$. At cryogenic temperatures especially in the ASE regime, $Q_0$ is far more sensitive to unanticipated perturbations in geometry than bulk material properties. The RRR values of 10 and 50, here, are then more correctly viewed as effective values used to modify our simulations of low-temperature performance as per the next section.

While the exact reason for the reduced value in CYBORG is not precisely known, there are two features to note. Based on the location of the seam for brazing necessary to machine the high shunt impedance nose cones, excess braze material around the cavity circumference was introduced. This had the effect of detuning the cavity by 9 MHz before cooling, from ≈5694 to ≈5703 MHz. In addition, the demountable backplane was only able to be tightened to achieve a room temperature $Q_0$ value of ≈7800 without further detuning the cavity as opposed to the design value of ≈8500. The coupling was also slightly undervalued at ≈0.6 instead of ≈0.7. As a result, the cooling process may be deforming the backplane in an unexpected manner affecting the FoM but most likely not the material properties themselves.

We note that also there is a difference between the normalized coupling $\beta$ and $Q_0$ in an unexpected manner. Since they both have the same dependence on electrical conductivity, their enhancement factors (when normalized to the room temperature value) should be equal as a function of temperature. Indeed, the definition of coupling is $\beta = \frac{Q_0}{Q_E}$ where $Q_E$ is the quality factor of the external circuit. $Q_E$, in theory, should have no explicit dependence on electrical conductivity. This implies that $Q_E$ is changing in the course of our RF antenna measurements. Qualitatively, a possible explanation here is that while the cryogenic shrinking of the coupling port is explicitly included in theory shown in Figure 5, the taper across the temperature gradient in the waveguide is not. This additional waveguide taper may be reducing the coupling as function of temperature and so reducing the enhancement factor. The thermal contraction of the waveguide will be incorporated in future simulations. Another possible explanation may be a systematic error associated with the long distance of the antenna to the gun. Future studies will include RF measurement through the dummy load port close to the gun.

Using the methodology established in the previous section, we can compute certain RF figures of merit for both CYBORG phases using the empirically measured values and the effective RRR curves. Otherwise, simulations using NIST data are reported using the

reported uncertainties. Values for the gun at certain relevant temperatures are shown in Table 2. Currently, the measurements are limited to low-power testing, but will be continually verified with high-power measurements as the klystron power is increased to the intended 1 MW. Using the empirical measurements, we can also see in Table 2 how much smaller we can make our uncertainties from direct measurements as opposed to NIST curves in simulation.

**Table 2.** Operational parameters for CYBORG at several different temperatures representing important working points. Values are from simulation where necessary and measurement when possible

| Parameter | 295 K | 95 K | 77 K | 45 K |
|---|---|---|---|---|
| $f_0$ [MHz] | $5703.6 \pm 0.1$ [1] | $5720.410 \pm 0.003$ [1] | $5721 \pm 3$ | $5722 \pm 4$ |
| $Q_0$ | $7808 \pm 13$ [1] | $14{,}326 \pm 12$ [1] | $21{,}000 \pm 3600$ | $30{,}000 \pm 9900$ |
| Coupling $\beta$ | $0.608 \pm 0.002$ [1] | $1.069 \pm 0.002$ [1] | $1.60 \pm 0.44$ | $2.4 \pm 0.9$ |
| Filling time [μs] | $0.271 \pm 0.01$ [1] | $0.386 \pm 0.001$ [1] | $0.44 \pm 0.01$ | $0.49 \pm 0.03$ |
| Power [MW] for 120 MV/m | $1.23 \pm 0.10$ | $0.85 \pm 0.08$ | $0.79 \pm 0.01$ | $0.70 \pm 0.09$ |
| Energy [J] per 2 μs pulse | $2.45 \pm 0.01$ | $1.70 \pm 0.02$ | $1.58 \pm 0.03$ | $1.40 \pm 0.19$ |
| Cathode field @ 0.5 MW | $77 \pm 3$ MV/m | $92 \pm 5$ MV/m | $93 \pm 3$ MV/m | $102 \pm 7$ MV/m |

[1] Values experimentally measured or computed directly from low-power measurements.

The exact choices of each temperature were explained in the methodology but are again clarified here: 295 K is room temperature, which is comfortable within the operational bandwidth of our C-band power; 95 K is the current coldest cooldown the gun has achieved as per the results shown in Figure 5; 77 K is liquid nitrogen temperature, representing a significant cryogenic milestone as well as the working point for most of our simulation optimizations; and 45 K is the low-temperature reach goal for the UCXFEL photoinjector and represents the temperature of sustainable 240 MV/m fields during breakdown tests. For Phase 1, where a simple Cu cathode is present, the 95 K working point is sufficient from completion. For Phase 2, which will include a low-temperature load lock allowing for the insertion of lower MTE higher QE semiconductor cathodes, the 77 K working point becomes the goal.

Before moving on to beam line GPT simulations, we note that, implicitly, they contain RF pulse characteristics from our commissioned C-band system. The pulses are flat top 2 μs with current commissioned peak power of 0.5 MW. From Table 2 we can see that even with the reduced $Q_0$ and $\beta$ we are still within tolerance to achieve 120 MV/m for Phase 1 using under 1 MW of RF power. Furthermore, our commissioned power will still provide a peak cathode field between 80–90 MV/m. Currently rep rate is limited by the thermal requirements of the photogun to 1 Hz.

*3.3. Beamline Status*

Combining the measurements established above, we can better inform the temperature-dependent quantities of the RF gun. From this information, we compute the field profile and beam dynamics emittance compensation given the available pulse with power from the real measured solenoid and the measured laser spectrum to inform our realistic beamline simulations. We have fabricated the near-term solenoid for use on the CYBORG beamline as specified by our Materials and Methods section. The measured field profiles are show in Figure 6 along with a photograph within the existing Phase 1 configuration. We examine a simplified beamline composed of gun section, quad doublet, in-vacuum mirror box, YAG screen, and Faraday cup. The beamline has been commissioned up to the solenoid

and first diagnostic screen. The solenoid has to be placed at a minimum cathode plane distance of 25 cm as it must be outside our gun cryostat. Due to additional practical implications, the YAG screen is placed further downstream at 60 cm. We measured our laser spectrum using an Ocean Optics spectrometer which after frequency doubling leads to an approximate Gaussian profile peaked at 265 nm.

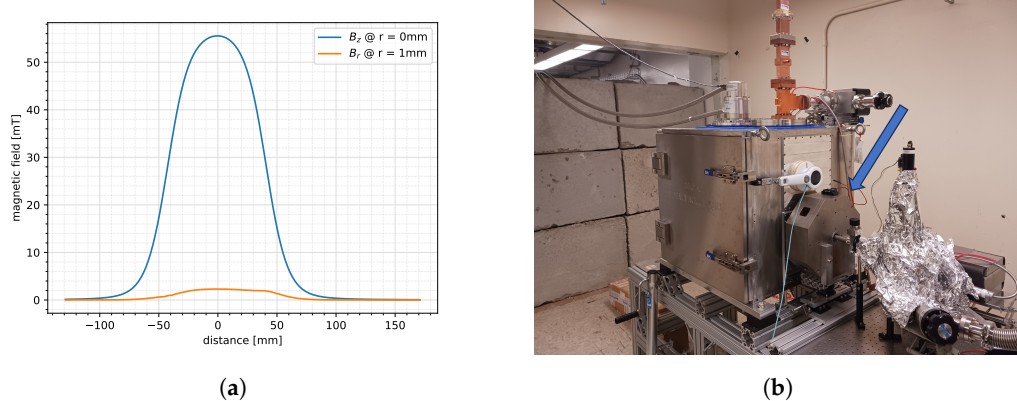

| (a) | (b) |

**Figure 6.** (**a**) Measured magnetic field profile to be used in beamline optimization simulations. On axis *B*-field and r component (at $r = 1$ mm), measurements of solenoid were made with our high-precision gantry. (**b**) Location of solenoid indicated by blue arrow in Phase 1 beamline.

## 4. Discussion

As Phase 1 of the CYBORG beamline approaches completion, Phase 2 studies are being ramped up. Within the context of existing cathode test beds, the parameters for both phases are included in Table 3. The Phase 2 gun configuration is shown in Figure 7, where it depicts the initial conception of a cryogenic capable load lock. The consideration include an interference fit coupling for the INFN style minipuck cathode insertion. A Molybdenum substrate inserted into the copper backplane would seal in the copper via differential coefficients of thermal expansion creating a strong mechanical and RF seal. Initial cool down tests of this have been promising [55]. Furthermore, we can more seriously develop a simulation suite for different cathodes of interest from a cryoemission standpoint [7,56–58].

**Table 3.** Parameters for CYBORG to place it within the context of the existing test beds shown in Table 1.

| Parameter | CYBORG Phase 1 | CYBORG Phase 2 |
| --- | --- | --- |
| Cavity type | NCRF | - |
| Cavity geometry | 0.5 cell re-entrant | - |
| Cathode assembly | Demountable Cu backplate | Cryogenic load lock |
| Design frequency | 5.712 GHz | 5.700–5.720 GHz |
| Peak cathode field | $\geq$120 MV/m | - |
| Operating temperature | 300–95 K [1] | 300–77 K |

[1] Current lowest temperature achieved with additional plans for 77 K operation.

Due to the very small emittance in the case of cryogenic temperatures as shown in Figure 1, highly sensitive diagnostics are required. To this end are currently working on optimizing the so-called TEM grid method which was developed and demonstrated at the PEGASUS beamline at UCLA and the SINBAD-ARES linac [59–61]. The general idea is to use an inexpensive metallic grid insert usually used for transmission electron microscopy (TEM), which can be held at a given potential instead of a traditional pepper pot mask. The TEM grid has significantly wider apertures losing less charge in beam measurement than alternative options. The technique is limited to 4D emittance measurements, so for full

reconstruction, additional bunch length measurement is necessary, for which an X-band deflecting cavity is planned. The deflecting cavity may also be useful as we consider future applications on CYBORG type guns to ultrafast electron diffraction and other applications, which require very small energy spread [7,8].

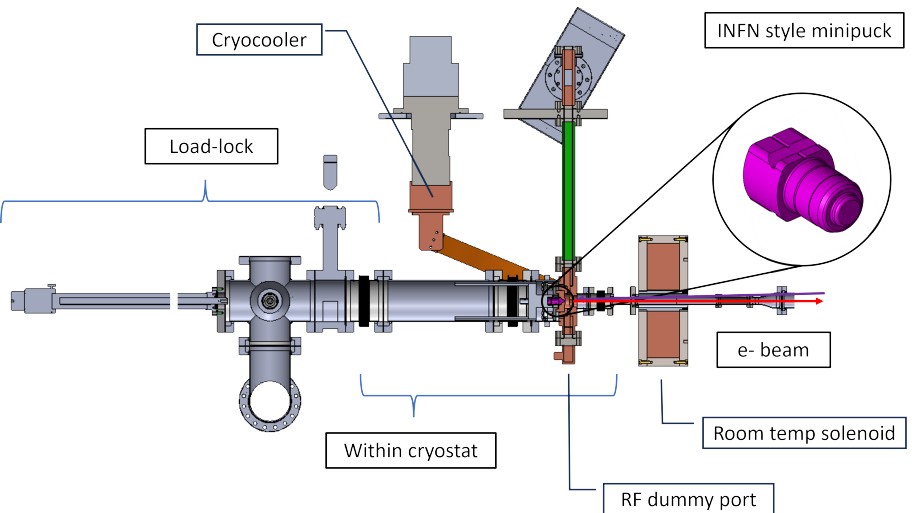

**Figure 7.** Phase 2 CYBORG cutaway showing INFN minipuck as an inset (in purple) and load-lock extension. Extension to thermal simulation are needed for further development.

Furthermore, in collaboration with SLAC and Los Alamos National Laboratories, a 1.6 cell photoinjector gun has been designed for room-temperature high-gradient operation using the same distributed coupling re-entrant optimization [20,62,63]. The next version of the UCXFEL photoinjector development will use the the same cavity design but modified for low-temperature operation. The further context of the new design is presented in the following [53].

## 5. Conclusions

The CYBORG beamline at UCLA is designed to use a new compact cryogenic C-band photogun to allow the study of advanced cathodes in a novel extreme cryogenic high-field environment. The device is currently nearing the completion of its first phase of development with promising results reported. The wide bandwidth and resonant modes of the gun cavity allow study in the range of room temperature down to 95 K with improved cooling capability planned down to 77 K in future iterations. The novel cavity geometry and low temperatures also offer RF stability and breakdown reduction, allowing high gradients in excess of 120 MV/m for less than 1 MW of power at 100 K. Currently only limited by the commissioned RF klystron power supply of 0.5 MW, we have access to fields in the range of 80–90 MV/m at 100 K with near-term improvements up to 1 MW to follow. This places the CYBORG beamline comfortably within a regime not accessible by other test beds. The lessons learned here are furthermore critical in informing the next generation of cryogenic ultra-high brightness photoinjectors and linear accelerator cavities.

**Author Contributions:** Conceptualization, G.E.L. and J.R.; methodology, G.E.L., N.M., Z.L., A.F., S.T. and J.R.; software, G.E.L., F.B. and M.C.; formal analysis, G.E.L. and F.B.; investigation, G.E.L., F.B., M.C., A.F. and M.Y.; resources, A.F., Y.S., S.T., O.W. and J.R.; writing—original draft preparation, G.E.L.; writing—review and editing, G.E.L. and J.R.; visualization, G.E.L.; supervision, J.R.; funding acquisition, J.R. All authors have read and agreed to the published version of the manuscript.

**Funding:** This work was supported by the Center for Bright Beams, National Science Foundation Grant No. PHY-1549132 and DOE HEP Grant DE-SC0009914.

**Data Availability Statement:** The raw data supporting the conclusions of this article will be made available by the authors on request.

**Conflicts of Interest:** The authors declare no conflicts of interest. The funders had no role in the design of the study; in the collection, analyses, or interpretation of data; in the writing of the manuscript; or in the decision to publish the results.

## Abbreviations

The following abbreviations are used in this manuscript:

| | |
|---|---|
| ASE | Anomolous skin effect |
| BDR | Breakdown rate |
| $C^3$ | Cool Copper Collider |
| CYBORG | CrYogenic Brightness-Optimized Radiofrequency Gun |
| DC | Direct current |
| FoM | Figures of merit |
| ILC | International Linear Collider |
| INFN | Istituto Nazionale di Fisica Nucleare |
| LCLS | Linear Coherant Light Source |
| MLI | Multi-layer insulation |
| MTE | Mean transverse energy |
| NC | Normal conducting |
| NIST | National Institute of Standards and Technologies |
| PFN | Pulse-forming network |
| QE | Quantum efficiency |
| RF | Radiofrequency |
| RRR | Residual resistivity ratio |
| SRF | Superconducting radiofrequency |
| UCLA | University of California, Los Angeles |
| UCXFEL | Ultra-compact Xray Free Electron Laser |

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
