# Peer review of "Improving Cathode Testing with a High-Gradient Cryogenic Normal Conducting RF Photogun"

_instruments, doi:10.3390/instruments8010014_

Round 1

Reviewer 1 Report

Comments and Suggestions for Authors

The work is interesting, given the framework of a conference proceeding. It would be interesting to know if the evaluations taken into account on thermal emittance are including the scaling with respect of the peak field on cathode. There are results not well in agreement with expectations and explanations could be deeper. 

Comments on the Quality of English Language

There are several typos, sentences that don't sound very smooth and missing parts (ref. to figures, captions of tables). It is clearly a work carried out with a limited effort (at least in the paper writing phase). I'm not going to review word by word as I'm used to do, given the amount of revisions needed in this case.

Reviewer 2 Report

Comments and Suggestions for Authors

Comments on the Quality of English Language

Reviewer 3 Report

Comments and Suggestions for Authors

Please ask the authors to read the paper and correct all the errors before we go into technical details. A few examples:

1, There is a list of abbreviations at the end of this paper, it did not cover all abbreviations in this paper, and also in the paper, some abbreviations were used at the beginning and then explained later, for example, UCXFEL in line 15 and line 39.

2, The captions for Table 2 and 3.

3, Line 62, Figure ??

4, Line 225, reference missing.

Comments on the Quality of English Language

No comment

Round 2

Reviewer 3 Report

Comments and Suggestions for Authors

1, Again, abbreviations. Please carefully check throughout the paper and make necessary corrections. Here is an incomplete list of errors:
   The following abbreviations were defined twice: breakdown rates (BDR), CrYogenic Brightness-Optimized Radiofrequency Gun (CYBORG)
   The following abbreviations were used before they were defined: MTE, RF
   The following abbreviations need to be defined: X-ray free electron lasers (XFEL), SRF
2, In the "Materials and Methods" section, it seems to me all the guns in Table 1 use Cu as the photocathode. Please clearly state this information. HZDR and BNL also have guns that use different photocathode, please briefly explain why you did not try to compare with them in table 1, any disadvantage of HZDR or BNL designs/photocathode in your application?
3, Fig. 3 was not mentioned in the paper. I believe it showed CST simulation results, it should be mentioned whether this E field plot is for 1 joule, or it is for certain kV operation voltage.
4, In line 110: "but they become more challenging when temperature dependence is introduced.", please briefly discuss the challenges. I understand the major concerns here are temperature dependence of the thermal conductivity (cooling) and surface resistance (RF loss, coupling), these should be clearly stated.
5, In line 164: "Progress is ongoing since the technology is quite advanced so for the near term we have manufactured a conventional coil wound solenoid around the optimized yoke geometry." Please briefly explain the challenges, what is the current state-of-art and what is your design.
6, It is mentioned "the RRR are in the range of several hundred" for cavity and in Fig. 5(b), RRR 10~50 is used, what is the RRR of the cavity Cu and why 10~50 is used here? Is there a surface quality degradation here? Please explain in line ~196.
7, In line 202, it should be explained that Q_E is the coupler's external Q. Also in Fig. 5(b), I believe "CYBORG Q" is Q_0. From the plot it is difficult to identify the change of ratio Q_0/beta = Q_E, it will be great if it can be plotted in Fig. 5(b).
8 In Fig. 5(a) and in the first paragraph of page 8, it is mentioned 45 K is the goal. Please briefly explain why there is a gap between current status and the goal, and what needs to be done to achieve the goal. Discussion needs to be made before you mention it in "Conclusions" section.
